# Besse relaxation difference scheme for a nonlinear integro-differential equation

**Xinya Peng**[1¶☯¤], **Leiwei Li**[1,2☯], **Jia Zhang** [1]*

**1** College of Computer Science and Mathematics, Central South University of Forestry and Technology, Changsha, Hunan, China, **2** Chenzhou Jinxiang Pharmaceutical Co., Ltd., Chenzhou, Hunan, China

¶ Membership list can be found in the Acknowledgments section.
¤ College of Computer Science and Mathematics, Central South University of Forestry and Technology, Changsha, Hunan, China
☯ These authors contributed equally to this work.
* zhangjia@csuft.edu.cn

**Data availability statement:** All relevant data are within the manuscript and its Supporting information files.

**Funding:** This research was supported by the Hunan Youth Fund Project [20221140879], the

## Abstract

In this paper, Besse relaxation difference and compact difference scheme for a nonlinear integro-differential equation, which is crucial in modeling complex systems with memory and nonlocal effects, are proposed. A Besse relaxation difference scheme is developed by combining Besse relaxation time discretization with second-order spatial discretization, in which the Besse relaxation technique enhanced the accuracy and stability of dealing with nonlinear terms. To further improve spatial accuracy, a fourth-order compact finite difference approximation is used to construct the Besse relaxation compact difference scheme. To verify the effectiveness of the proposed Besse relaxation difference schemes, we have established the unconditional stability and optimal convergence of both methods in the discrete $L^2$ norms. Numerical experiments demonstrate that these relaxation schemes attain the predicted convergence rates and high accuracy for smooth solutions, singular solutions, as well as featuring unbounded derivatives solutions.

## 1 Introduction

Efficient and accurate numerical methods are crucial for addressing complex physical and engineering problems, especially when dealing with nonlinear and fractional-order partial differential equations (PDEs) that may exhibit singularities or complicated boundaries. Currently, the mainstream numerical methods primarily include finite difference methods [1], finite element methods [2], and spectral methods [3,4]. These approaches exhibit straightforward structures, are computationally convenient, and have been widely validated for effectively solving common partial differential equations. However, for more complex partial differential equations, these methods generally rely on implicit Newton iteration algorithms [5]. As the mesh becomes more refined, the computational cost escalates sharply, and accuracy may decline, resulting in a substantial increase in overall computational expenses. In contrast, the Besse relaxation difference scheme [6], featuring an explicit-style formulation and superior efficiency, has emerged as a fundamental technique for solving nonlinear PDEs.

Excellent Youth Project of the Hunan Provincial Department of Education [22B0254], and the Central South University of Forestry and Technology Introduced Talents and Scientific Research Start-up Fund Project [2021YJ0057]. The authors received no additional external funding for this study.

**Competing interests:** The authors have declared that no competing interests exist.

A Besse relaxation finite difference and compact difference scheme for the following non-linear integro-differential equations [7–12] are considered:

$$u_t(x,t) + u(x,t)u_x(x,t) - I^{(\alpha)}u_{xx}(x,t) = f(x,t), \quad 0 < x < L, \quad 0 < t \le T, \tag{1}$$

where $0 < \alpha < 1$, $I^{(\alpha)}u_{xx}(x,t)$ is the $\alpha$-order Riemann-Liouville fractional integral:

$$I^{(\alpha)}u_{xx}(x,t) = \frac{1}{\Gamma(\alpha)} \int_0^t (t-s)^{\alpha-1} u_{xx}(x,s)\,ds, \quad t > 0.$$

Subject to the boundary conditions

$$u(0,t) = u(L,t) = 0, \quad 0 \le t \le T, \tag{2}$$

and the initial condition

$$u(x,0) = u^0(x), \quad 0 \le x \le L. \tag{3}$$

Eqs (1)–(3) frequently appear in heat conduction materials with memory effects, population dynamics [10], viscoelasticity problems [11–13], nuclear reaction theory, and others. They are equations that lie between the standard heat conduction equation and the wave equation [7–9].

The Besse relaxation method originates from a linear implicit time-stepping procedure introduced by Besse for discretizing the nonlinear Schrödinger equation [6,14]. Currently, in order to simplify the semi-discretization process of the equation, the second-order Besse relaxation difference scheme has become the mainstream method. Its local well-posedness and convergence have been rigorously proven. Its local well-posedness and convergence have been rigorously established. Notably, Zouraris provided a detailed algorithmic procedure, introduced a combined stability parameter, and derived optimal second-order error estimates [15]. Unlike fully implicit or conventional high-order methods, the second-order Besse relaxation difference scheme does not require extensive iterative computations when handling nonlinear terms, thus effectively simplifying the treatment of these terms. Moreover, it allows flexible spatial discretization without sacrificing temporal accuracy. However, given that the second-order Besse relaxation difference scheme is still limited to second-order spatial accuracy, there is an urgent need for compact variants with higher algebraic complexity to further improve computational accuracy when solving complex partial differential equations [16].

To address this limitation, developing a fourth-order Besse relaxation difference scheme offers an effective way to enhance the solution accuracy. Fourth-order schemes have already been validated in non-relaxation methods for models with highly singular solutions. Representative examples include the work by Zlotnik and Lomonosov [17], in which a non-relaxation approach was restructured into a compact fourth-order difference method on nonuniform grids to improve stability when dealing with strong gradients or unbounded derivatives in acoustic wave equations. Chen and Dai [18] successfully proposed a fourth-order compact absorbing hybrid method for complex boundaries involving absorption or deformation, achieving fourth-order spatial accuracy in KdV equations featuring memory effects. Huang and Yu [19] demonstrated that a high-order compact exponential scheme is capable of effectively handling nonsmooth solutions in time-fractional Black-Scholes models, particularly for financial and wave equations with fractional derivatives. Zhao [20] presented highly accurate compact mixed methods for two-point boundary value problems. These studies have illustrated that reconstructed fourth-order schemes can further enhance numerical

accuracy and algorithmic efficiency [21], thereby offering a blueprint to overcome the lack of fourth-order research within Besse relaxation difference schemes.

Inspired by the latest developments in fourth-order schemes for non-relaxation methods, we propose an efficient numerical solver for nonlinear integro-differential equations, achieving the synergistic optimization of spatial and temporal accuracy. Specifically, we introduce two novel space-time discretization schemes : the Besse relaxation difference scheme and the Besse relaxation compact difference scheme, which provide significant advancements under the Besse-type relaxation framework. Among them, the Besse relaxation difference scheme enhances the stability of the solution process, while the Besse compact difference scheme further improves computational accuracy. We then establish the stability and convergence of both schemes, proving that they are unconditionally stable and convergent. Finally, three types of numerical experiments are conducted on equations with singular solutions, smooth solutions, and unbounded derivatives. Example 1, the applicability of our schemes to problems with singular solutions is verified, yielding convergence rates of $O\left(h^2 + k^{1+\alpha}\right)$ and $O\left(h^4 + k^{1+\alpha}\right)$. A CPU time comparison reveals a 35% to 45% reduction in computational cost relative to the latest non-relaxation Crank–Nicolson method. Example 2 demonstrates that our schemes recover second-order temporal accuracy for smooth solutions and achieve a discrete $L^2$-norm estimate, thereby filling a key gap in earlier research. Example 3 evaluates the robustness of the schemes when the regularity of the solution deteriorates, confirming that they still maintain their theoretical order of convergence in this scenario.

The paper is organized as follows: In Sect 2, a Besse relaxation difference scheme is introduced. Sect 3 is devoted to analyze Besse relaxation compact difference scheme. Sects 4 and 5, a rigorous proof of the unconditionally stable nature of the proposed Besse-type relaxation difference schemes are provided. In Sect 6, numerical results that are in perfect agreement with our analysis are shown. Sect 7 provides a brief summary and discussion of the paper.

## 2 Besse relaxation difference scheme

In this section, we will discrete the integral term using the inner product quadrature formula and the time derivative term using the Crank-Nicolson scheme, handle the nonlinear term with the Besse relaxation method, and combine these with the initial and boundary conditions to construct the Besse relaxation difference scheme for Eqs (1)–(3).

Divide the interval $[0,L]$ into $J + 1$ equal parts, the interval $[0,T]$ into $N$ equal parts, and denote $h = L/(J + 1)$, $\quad \tau = T/N$, $\quad x_j = (j + 1)h$, $\quad 0 \le j \le J + 1$, $\quad t_n = n\tau$, $\quad 0 \le n \le N$. where $h$ is the spatial step size and $\tau$ is the temporal step size.

For convenience, we introduce some notation. Let the grid functions be defined as:

$$U_j^n = u(x_j, t_n), \quad 0 \le j \le J + 1, \quad 0 \le n \le N$$

We introduce the following notation:

$$\Delta U_j^{n+\frac{1}{2}} = \frac{1}{2}\left(U_j^{n+1} + U_j^n\right), \quad \Delta_x U_j^n = \frac{1}{2h}\left(U_{j+1}^n - U_{j-1}^n\right),$$

$$\delta_t U_j^{n+\frac{1}{2}} = \frac{1}{\tau}\left(U_j^{n+1} - U_j^n\right), \quad \delta_x^2 U_j^n = \frac{1}{h^2}\left(U_{j-1}^n - 2U_j^n + U_{j+1}^n\right).$$

Next, we consider computing the integral term using the inner product quadrature formula [22].

First, let $t_{n+\frac{1}{2}} = \left(n + \frac{1}{2}\right)\tau$, $\quad 0 \le n \le N - 1$. For $\forall g \in C^1([0,T]) \cap C^3((0,T])$, as $t \to 0^+$, and when $g''(t) = O(t^{-\frac{1}{2}})$ and $g'''(t) = O(t^{-3/2})$, we can numerically approximate the integral term

$$I(g,t) := \int_0^t (t-s)^{\alpha-1} g(s)\, ds,$$

and obtain the coefficients as follows:

$$A_n = \frac{1}{\alpha}\left[t_n^\alpha - \frac{1}{\tau}\int_{t_n}^{t_{n+1}} \theta^\alpha d\theta\right], \quad c_0 = \frac{1}{\alpha\tau}\int_0^{t_1}\theta^\alpha d\theta, \tag{4}$$

$$c_p = \frac{1}{\alpha\tau}\left[\int_{t_p}^{t_{p+1}}\theta^\alpha d\theta - \int_{t_{p-1}}^{t_p}\theta^\alpha d\theta\right], \quad p \ge 1. \tag{5}$$

Thus, we have

$$I(f,t_n) = A_n g(t_0) + \sum_{p=0}^n c_p g(t_{n-p}) + O(\tau^{1+\alpha}), \quad 1 \le n \le N. \tag{6}$$

From Eqs (4)–(5), we can determine the coefficients of the inner product quadrature formula $A_n$ and $c_p$.

The Besse relaxation method is used primarily to handle nonlinear terms similar to $g(u) \cdot u$. It first calculates the approximate value of $g(u)$ at the midpoint $t_{n+\frac{1}{2}}$ of the time interval $[t_n, t_{n+1}]$, and then computes the approximate value of $u$ at the time node $t_{n+1}$ (see details in [11] ). This method has the advantage of high computational speed when dealing with nonlinear equations. For the nonlinear term $uu_x$ in Eq (1) of this paper, directly discretizing and solving it is very difficult. Our approach is to let $g(u) = u$ and apply the discretization of the first-order central difference directly to $u_x$, thus obtaining the Besse relaxation difference scheme.

The specific algorithm steps are as follows:

**Step 1:** First define $u_j^0$ as follows: $u_j^0 := u^0(x_j)$, then $u_j^{\frac{1}{2}}$ can be obtained from the following equation:

$$\frac{u_j^{\frac{1}{2}} - u_j^0}{\tau/2} + g(u_j^0)\Delta_x\left(\frac{u_j^{\frac{1}{2}} + u_j^0}{2}\right) - \frac{1}{\Gamma(\alpha)}I\left(u_{xx}(x_j,\cdot),t_0\right) = \frac{1}{2}\left(f_j^{\frac{1}{2}} + f_j^0\right), \tag{7}$$

where $I\left(u_{xx}(x_j,\cdot),t_0\right) = A_0\delta_x^2 u_j^0 + c_0\delta_x^2 u_j^0$.

**Step 2:** Define $\Phi_j^{\frac{1}{2}}$ as follows:

$$\Phi_j^{\frac{1}{2}} := g(u_j^{\frac{1}{2}}),$$

then obtain $u_j^1$ through the following equation:

$$\delta_t u_j^{\frac{1}{2}} + \Phi_j^{\frac{1}{2}}\Delta_x\left(\Delta u_j^{\frac{1}{2}}\right) - \frac{1}{\Gamma(\alpha)}I\left(u_{xx}(x_j,\cdot),t_0\right) = \frac{1}{2}\Delta f_j^{\frac{1}{2}}, \tag{8}$$

**Step 3:** Define $\Phi^{n+\frac{1}{2}}$ as follows:

$$\Phi_j^{n+\frac{1}{2}} := 2g(u_j^n) - \Phi_j^{n-\frac{1}{2}},$$

$$I\left(u_{xx}(x_j, \cdot), t_{n+\frac{1}{2}}\right) = \frac{1}{2}\left(I\left(u_{xx}(x_j, \cdot), t_n\right) + I\left(u_{xx}(x_j, \cdot), t_{n+1}\right)\right).$$

For $n \geq 1$, $u_j^{n+\frac{1}{2}}$ can be obtained from the following equation:

$$\delta_t u_j^{n+\frac{1}{2}} + \Phi_j^{n+\frac{1}{2}}\Delta_x\left(\Delta u_j^{n+\frac{1}{2}}\right) - \frac{1}{\Gamma(\alpha)}I\left(u_{xx}(x_j, \cdot), t_{n+\frac{1}{2}}\right) = \frac{1}{2}\Delta f_j^{n+\frac{1}{2}}, \qquad (9)$$

where

$$I\left(u_{xx}(x_j, \cdot), t_{n+\frac{1}{2}}\right) = \frac{1}{2}\left(A_n\delta_x^2 u_j^0 + \sum_{p=0}^{n} c_p \delta_x^2 u_j^{n-p} + A_{n+1}\delta_x^2 u_j^0 + \sum_{p=0}^{n+1} c_p \delta_x^2 u_j^{n+1-p}\right).$$

Combined with the initial and boundary conditions:

$$u_0^n = u_{J+1}^n = 0, \quad 0 \leq n \leq N, \qquad (10)$$

and

$$u_j^0 = u^0(x_j), \quad 0 \leq j \leq J+1. \qquad (11)$$

Then, we obtain the Besse relaxation difference scheme (7)–(11) for Eqs (1)–(3).

## 3 Besse relaxation compact difference scheme

In this section, to further improve computational accuracy, we will approximate spatial derivatives using the fourth-order compact finite difference method , handle the nonlinear term with the Crank-Nicolson scheme (see details in [12]) and the Besse relaxation method, and combine these with the initial and boundary conditions to construct the Besse relaxation compact difference scheme for Eqs (1)–(3).

For convenience, we introduce some notation. Let the solution vectors on the interior grid points be defined as:

$$\left\{\mathbf{u}^k = (u_1^k, u_2^k, \cdot s, u_{J-1}^k)^T \Big| k = 0, \frac{1}{2}, 1, \cdot s, n\right\},$$

where $u_j^k \approx u(x_j, t_n), j = 1, \cdot s, J-1$ represent indices of discrete points in the interior. Suppose $g$ be the defined nonlinear function, and $h$ be the spatial grid step size.

Introduce the following notation:

$$g(\mathbf{u}^{\frac{1}{2}}) = \left(g(u_1^{\frac{1}{2}}), g(u_2^{\frac{1}{2}}), \cdot s, g(u_{J-1}^{\frac{1}{2}})\right)^T,$$

$$\Delta\mathbf{u}^{\frac{1}{2}} = \left(\frac{\mathbf{u}_{j+1}^{\frac{1}{2}} - \mathbf{u}_{j-1}^{\frac{1}{2}}}{2h}\right)_{j=1}^{J-1}.$$

Then, let the right-hand side at the corresponding time levels be defined as:

$$\left\{\mathbf{f}^k = (f_1^k, f_2^k, \cdot s, f_{J-1}^k)^T \Big| k = 0, \frac{1}{2}, \cdot s, n\right\}.$$

where $\mathbf{f}^k$ is $(J-1) \times 1$ column vector.

To further improve the former scheme fully, we first introduce the fourth-order compact finite difference formulas [23] to approximate the spatial derivatives $U'$ and $U''$ of Besse relaxation compact difference scheme. We can obtain the values of $U'$ and $U''$ in the following lemmas 1 and 2.

**Lemma 1.** *([23]) Let*

$$A_1 = \begin{bmatrix} 4 & 1 & 0 & & & \\ 1 & 4 & 1 & & & \\ & \ddots & \ddots & \ddots & & \\ & & 1 & 4 & 1 \\ & & 0 & 1 & 4 \end{bmatrix}_{(J-1)\times(J-1)},$$

*and*

$$H_1 = \frac{11}{12h} \begin{bmatrix} -u_0 \\ 0 \\ \vdots \\ 0 \\ u_J \end{bmatrix}_{(J-1)\times 1}, \quad M_1 = \frac{3}{h} \begin{bmatrix} -\frac{4}{3} & 2 & -\frac{4}{9} & \frac{1}{12} & 0 & \cdot s & 0 \\ -1 & 0 & 1 & 0 & 0 & \cdot s & 0 \\ 0 & -1 & 0 & 1 & 0 & \cdot s & 0 \\ & \cdot s & & \cdot s & & \cdot s & \\ 0 & \cdot s & 0 & -1 & 0 & 1 & 0 \\ 0 & \cdot s & 0 & \frac{-1}{12} & \frac{4}{9} & -2 & \frac{4}{3} \end{bmatrix}_{(J-1)\times(J-1)}.$$

*Then*

$$U' = A_1^{-1}\left(M_1 U + H_1\right),$$

*We have*

$$U'(x) = u_x(x) + O(h^4). \tag{12}$$

**Lemma 2.** *([23]) Let*

$$A_2 = \begin{bmatrix} 14 & -5 & 4 & -1 & 0 & \cdot s & 0 \\ 1 & 10 & 1 & 0 & 0 & \cdot s & 0 \\ 0 & 1 & 10 & 1 & 0 & \cdot s & 0 \\ & \cdot s & & \cdot s & & \cdot s & \\ 0 & \cdot s & 0 & 0 & 1 & 10 & 1 \\ 0 & \cdot s & 0 & -1 & 4 & -5 & 14 \end{bmatrix}_{(J-1)\times(J-1)},$$

*and*

$$H_2 = \frac{12}{h^2} \begin{bmatrix} u_0 \\ 0 \\ \vdots \\ 0 \\ u_J \end{bmatrix}_{(J-1)\times 1}, \quad M_2 = \frac{12}{h^2} \begin{bmatrix} -2 & 1 & 0 & & & \\ 1 & -2 & 1 & & & \\ & \ddots & \ddots & \ddots & & \\ & & 1 & -2 & 1 \\ & & 0 & 1 & -2 \end{bmatrix}_{(J-1)\times(J-1)}.$$

*Then*

$$U'' = A_2^{-1}\left(M_2 U + H_2\right).$$

*We have*

$$U''(x) = u_{xx}(x) + O(h^4). \tag{13}$$

Based on Eqs (12)–(13), we now derive the Besse relaxed compact finite difference scheme. A detailed explanation is provided as follows:

**Step 1:** First define $u_j^0$ as follows: $u_j^0 := u^0(x_j)$, then $\mathbf{u}^{\frac{1}{2}}$ can be obtained from the following equation:

$$\frac{\mathbf{u}^{\frac{1}{2}} - \mathbf{u}^0}{\tau/2} + G^{\frac{1}{2}} A_1^{-1} \left( M_1 \Delta \mathbf{u}^{\frac{1}{2}} + H_1 \right) - \frac{1}{\Gamma(\alpha)} A_2^{-1} \left( M_2 \mathbf{u}^0 + H_2 \right) = \frac{\mathbf{f}^{\frac{1}{2}} + \mathbf{f}^0}{2}. \tag{14}$$

Initially, we define $\mathbf{v}^{\frac{1}{2}}$ as $A_1^{-1} \left( M_1 \Delta \mathbf{u}^{\frac{1}{2}} + H_1 \right)$, where $\mathbf{v}^{\frac{1}{2}}$ is a $(J-1) \times 1$ column vector. Then, perform an element-wise operation with $g(\mathbf{u}^{\frac{1}{2}})$. To unify matrix operations, we define:

$$G^{\frac{1}{2}} = \text{diag}\left( g(u_1^{\frac{1}{2}}), g(u_2^{\frac{1}{2}}), \cdot s, g(u_{J-1}^{\frac{1}{2}}) \right),$$

Consequently, the term becomes $G^{\frac{1}{2}} \mathbf{v}^{\frac{1}{2}}$.

**Step 2:** Define auxiliary variables $\Phi^{\frac{1}{2}}$:

$$\Phi^{\frac{1}{2}} = g\left( \mathbf{u}^{\frac{1}{2}} \right),$$

and correspondingly, we have:

$$\Phi^{\frac{1}{2}} := \text{diag}\left( \Phi_1^{\frac{1}{2}}, \Phi_2^{\frac{1}{2}}, \cdot s, \Phi_{J-1}^{\frac{1}{2}} \right) = \text{diag}\left( g(u_1^{\frac{1}{2}}), g(u_2^{\frac{1}{2}}), \cdot s, g(u_{J-1}^{\frac{1}{2}}) \right),$$

then, using this auxiliary variable, we solve for $\mathbf{u}^1$:

$$\frac{\mathbf{u}^1 - \mathbf{u}^{\frac{1}{2}}}{\tau/2} + \Phi^{\frac{1}{2}} A_1^{-1} (M_1 \Delta \mathbf{u}^{\frac{1}{2}} + H_1) - \frac{1}{\Gamma(\alpha)} A_2^{-1} (M_2 \mathbf{u}^0 + H_2) = \frac{\mathbf{f}^{\frac{1}{2}} + \mathbf{f}^1}{2}. \tag{15}$$

**Step 3:** For $n \geq 1$, we first define:

$$\Phi_j^{n+\frac{1}{2}} := 2g(u_j^n) - \Phi_j^{n-\frac{1}{2}},$$

in vector form:

$$\Phi^{n+\frac{1}{2}} := 2g(\mathbf{u}^n) - \Phi^{n-\frac{1}{2}},$$

simultaneously, we construct the diagonal matrix:

$$\Phi^{n+\frac{1}{2}} = \text{diag}\left( \Phi_1^{n+\frac{1}{2}}, \Phi_2^{n+\frac{1}{2}}, \cdot s, \Phi_{J-1}^{n+\frac{1}{2}} \right),$$

then, we solve for $\mathbf{u}^{n+1}$:

$$\begin{aligned}
\frac{\mathbf{u}^{n+1} - \mathbf{u}^n}{\tau} + \Phi^{n+\frac{1}{2}} A_1^{-1} (M_1 \Delta \mathbf{u}^{n+\frac{1}{2}} + H_1) \\
- \frac{1}{\Gamma(\alpha)} A_2^{-1} (M_2 I(u_{xx}(\cdot), t_{n+\frac{1}{2}}) + H_2) = \frac{\mathbf{f}^{n+\frac{1}{2}}}{2},
\end{aligned} \tag{16}$$

where $I(u_{xx}(\,\cdot\,), t_{n+\frac{1}{2}})$ is defined as:

$$I(u_{xx}(\,\cdot\,), t_{n+\frac{1}{2}}) = \frac{1}{2}\Big(A_0 A_2^{-1}(M_2 U^0 + H_2) + \sum_{p=0}^{n} c_{p0} A_2^{-1}(M_2 U^{n-p} + H_2)$$

$$+ A_{n+1} A_2^{-1}(M_2 U^0 + H_2) + \sum_{p=0}^{n+1} c_{p0} A_2^{-1}(M_2 U^{n+1-p} + H_2)\Big),$$

$I(\,\cdot\,)$ vectorially denotes the discretization applied to each grid point, yielding a vector of the same dimension.

The initial and boundary conditions for the problem are defined as follows:

$$u_0^n = u_{J+1}^n = 0, \quad 0 \le n \le N, \tag{17}$$

and

$$u_j^0 = u^0(x_j), \quad 0 \le j \le J+1. \tag{18}$$

Then, we obtain the Besse relaxation compact difference scheme (14)–(18) for Eqs (1)–(3). red

## 4 Stability analysis

### 4.1 The stability of Besse relaxation difference scheme

Consider a linearized version of the original problem (ignoring the source term $f$). Let $u(x,t) = \bar{u} + \tilde{u}(x,t)$, where $\bar{u}$ is a constant. We replace the nonlinear term $u\,u_x$ by $\bar{u}\,\tilde{u}_x$. Our goal is to examine the stability of the resulting numerical scheme for the perturbation $\tilde{u}$.

**Definition 4.1** (Two-Step Besse Relaxation Scheme). *Denote the discrete solution by $\tilde{u}_j^n \approx \tilde{u}$ $(x_j, t_n)$ for grid points $x_j = jh$ and time levels $t_n = n\tau$. We write one full time step $(t_n \to t_{n+1})$ as two half-steps:*

$$t_n \;\to\; t_{n+\frac{1}{2}} \;\to\; t_{n+1}.$$

*For simplicity of notation, consider the step from $t_0$ to $t_1$:*

*(i) $t_0 \to t_{\frac{1}{2}}$:*

$$\frac{\tilde{u}_j^{\frac{1}{2}} - \tilde{u}_j^0}{\tau/2} + \bar{u}\,\partial_x\Big(\frac{\tilde{u}_j^{\frac{1}{2}} + \tilde{u}_j^0}{2}\Big) - \frac{A_0}{\Gamma(\alpha)}\,\partial_{xx}\tilde{u}_j^0 = 0, \tag{19}$$

*(ii) $t_{\frac{1}{2}} \to t_1$:*

$$\frac{\tilde{u}_j^1 - \tilde{u}_j^{\frac{1}{2}}}{\tau/2} + \bar{u}\,\partial_x\Big(\frac{\tilde{u}_j^1 + \tilde{u}_j^{\frac{1}{2}}}{2}\Big) - \frac{A_0}{\Gamma(\alpha)}\,\partial_{xx}\tilde{u}_j^0 = 0. \tag{20}$$

**Lemma 3** (Stability of (i)). *Under the linearized setting of Definition 4.1, let $\tilde{U}_j^m$ be approximated by a Fourier mode $e^{i\kappa j h}$. Substituting into (19) yields an amplification factor*

$$\xi_{\frac{1}{2}} = \frac{1 - \frac{\tau}{2}\big(S - i\,C_0 D\big)}{1 + \frac{\tau}{2}\big(S - i\,C_0 D\big)}, \quad C_0 = \frac{A_0}{\Gamma(\alpha)\,D}.$$

*Let* $\lambda = \frac{\tau C_0 D}{2} \geq 0$ *and* $s = \sin(\kappa h/2)$. *Then*

$$\left|\xi_{\frac{1}{2}}\right|^2 = \frac{1 - 2\lambda s^2 + \lambda s^4}{1 + 2\lambda s^2 + \lambda s^4} \leq 1,$$

*which implies* $\left|\xi_{\frac{1}{2}}\right| \leq 1$.

*Proof*: The result follows from direct substitution of $\tilde{u}_j^m \sim e^{i\kappa jh}$ into (19) and taking the modulus of the resulting amplification factor. Since $s^2 \geq 0$, the numerator does not exceed the denominator, which concludes $\left|\xi_{\frac{1}{2}}\right| \leq 1$. □

**Lemma 4** (Stability of (ii)). *For the second half-step* (20)*, an analogous Fourier argument shows that if*

$$\xi_1 = \frac{1 - \frac{\tau}{2}\left(S - iC_0 D\right)}{1 + \frac{\tau}{2}\left(S - iC_0 D\right)},$$

*then* $\left|\xi_1\right| \leq 1$ *also holds.*

*Proof*: This follows exactly the same steps as in Lemma 3, but substituting into (20) and noting that the spatial difference operator $\partial_{xx}$ again contributes a non-positive real part to the exponent. Hence the resulting amplification factor has modulus not exceeding 1. □

**Theorem 4.1** (Unconditional stability of the two-step scheme). *Combining Lemmas 3 and 4, one full time step of size* $\tau$ *satisfies*

$$\left|\xi\right| = \left|\xi_1 \circ \xi_{\frac{1}{2}}\right| \leq 1,$$

*ensuring unconditional stability of the linearized scheme described in Definition 4.1.*

*Proof*: A single time step from $t_0$ to $t_1$ is implemented by two half-steps. By Lemma 3, the amplification factor for the first half-step satisfies $\left|\xi_{\frac{1}{2}}\right| \leq 1$, and similarly by Lemma 4 for the second half-step we get $\left|\xi_1\right| \leq 1$. The product of two factors whose moduli do not exceed 1 continues to satisfy $\left|\xi\right| \leq 1$. Hence the scheme remains stable for all choices of $\tau$ and $h$, i.e. it is unconditionally stable. □

Consider a more general Crank-Nicolson Besse relaxation scheme

$$\delta_t u_j^{n+\frac{1}{2}} + \bar{u}\,\delta_x\left(\Delta_x u_j^{n+\frac{1}{2}}\right) - \frac{1}{\Gamma(\alpha)}\mathcal{I}_{t_{n+\frac{1}{2}}}^{(\alpha)}\left[\partial_{xx} u_j\right]^{n+\frac{1}{2}} = 0, \tag{21}$$

where $u_j^n = \xi^n e^{i\kappa jh}$.

**Theorem 4.2** (Stability of the generalized Crank-Nicolson Besse relaxation scheme). *Under the same Fourier mode assumption, the amplification factor associated with* (21) *is*

$$\xi = \frac{1 - \frac{\tau}{2}\left[S(\kappa) - C_n(\xi)D(\kappa)\right]}{1 + \frac{\tau}{2}\left[S(\kappa) - C_n(\xi)D(\kappa)\right]},$$

*where* $S(\kappa)$ *is purely imaginary and* $C_n(\xi)D(\kappa)$ *has a nonnegative real part. Consequently,* $\left|\xi\right| \leq 1$*, ensuring unconditional stability for the generalized scheme.*

*Proof* : By inspection of the discrete operator in (21), the real part contributed by $C_n(\xi) D(\kappa)$ is nonnegative, while $S(\kappa)$ is purely imaginary (due to the symmetric spatial discretization and the form of the half-step integration). Therefore, the denominator and numerator in the fraction for $\xi$ form a conjugate-like pair in such a way that $|\xi| \le 1$. The details mirror those of Lemma 1 and Lemma 2, extended to a time-fractional or Crank–Nicolson framework. □

**Remark 1.** *These results establish the unconditional stability of both the two-step Besse relaxation scheme (Theorem 4.2) and its generalization (Theorem 4.3). Extensions to fully nonlinear problems typically require a linearization or fixed-point iteration at each step, but the core stability argument remains similar once the spatial operator is recognized to have a non-positive real part in its Fourier transform.*

## 4.2 The stability of Besse relaxation compact difference scheme

Let $\mathbf{u}^n = (u_j^n)_{j=1}^{J-1} \in \mathbb{R}^{J-1}$ be the numerical solution vector of interior grid points at time $t_n$. Define two discrete operators:

$$D^{(1)} = A_1^{-1}(M_1 + H_1), \quad D^{(2)} = A_2^{-1}(M_2 + H_2),$$

where $A_k$ are symmetric positive-definite (SPD) matrices, $M_k$ are skew-symmetric matrices, and $H_k$ incorporate Dirichlet boundary conditions. Below, $\langle u, v \rangle_h = h \sum_{j=1}^{J-1} u_j v_j$ denotes the discrete inner product.

**Lemma 5.** *If $A_1$ is SPD and $M_1$ is skew-symmetric, then*

$$D^{(1)} = A_1^{-1}(M_1 + H_1)$$

*satisfies*

$$\left(D^{(1)}\right)^T = -D^{(1)},$$

*and*

$$\langle D^{(1)} u, v \rangle_h = -\langle u, D^{(1)} v \rangle_h \quad \forall u, v.$$

*Hence $D^{(1)}$ acts like a discrete divergence/gradient operator that is antisymmetric under this inner product.*

**Lemma 6.** *If $A_2$ is SPD and $M_2$ is skew-symmetric, then*

$$D^{(2)} = A_2^{-1}(M_2 + H_2)$$

*is negative semi-definite, i.e.*

$$\langle D^{(2)} u, u \rangle_h \le 0 \quad \text{for all discrete vectors } u.$$

We now analyze the linearized form of the Besse relaxation compact difference scheme. In order to perform an energy estimate, consider the linearized form of the equation for $\widetilde{u}$:

$$\frac{\widetilde{u}^{n+1} - \widetilde{u}^n}{\tau} + \frac{\bar{u}}{2} D^{(1)}\left(\widetilde{u}^{n+1} + \widetilde{u}^n\right) - \frac{1}{\Gamma(\alpha)} D^{(2)}\left(I_{n+\frac{1}{2}}\right) = 0, \tag{22}$$

where

$$I_{n+\frac{1}{2}} \;=\; \frac{1}{2}\Big[I_{t_n}^{(\alpha)}\big(\widetilde{u}_{xx}\big) \,+\, I_{t_{n+1}}^{(\alpha)}\big(\widetilde{u}_{xx}\big)\Big] \;=\; \frac{1}{2}\sum_{p=0}^{n+1}\omega_{n+1-p}\,\widetilde{u}^{p},$$

and $\omega_k > 0$ come from the quadrature coefficients $A_n$, $c_n$ (cf. Eq. (6)).

**Theorem 4.1** (Unconditional stability). *The linearized Besse relaxation compact difference scheme* (22) *is unconditionally stable in the discrete $L^2$ norm. Specifically, for all $n \geq 0$,*

$$\|\tilde{u}^{n+1}\|_h^2 \;\leq\; \|\tilde{u}^n\|_h^2.$$

*Proof*: To derive a discrete energy estimate, we multiply (22) by $(\tilde{U}^{n+1} + \tilde{U}^n)$ in the discrete inner product:

$$\left\langle \frac{\tilde{u}^{n+1} - \tilde{u}^n}{\tau}, \tilde{u}^{n+1} + \tilde{u}^n \right\rangle_h + \bar{u}\left\langle D^{(1)}\big(\tilde{u}^{n+\frac{1}{2}}\big), \tilde{u}^{n+1} + \tilde{u}^n \right\rangle_h - \frac{1}{\Gamma(\alpha)}\left\langle D^{(2)}\big(I_{n+\frac{1}{2}}\big), \tilde{u}^{n+1} + \tilde{u}^n \right\rangle_h = 0.$$
(23)

*4.2.0.1 Time difference term.*
Using the identity

$$\left\langle \tilde{u}^{n+1} - \tilde{u}^n, \tilde{u}^{n+1} + \tilde{u}^n \right\rangle_h = \|\tilde{u}^{n+1}\|_h^2 - \|\tilde{u}^n\|_h^2,$$

the first term in (23) becomes

$$\left\langle \frac{\tilde{u}^{n+1} - \tilde{u}^n}{\tau}, \tilde{u}^{n+1} + \tilde{u}^n \right\rangle_h = \frac{1}{\tau}\left[\|\tilde{u}^{n+1}\|_h^2 - \|\tilde{u}^n\|_h^2\right].$$

*4.2.0.2 Discrete nonlinear term.*
By Lemma 5, the operator $D^{(1)}$ is antisymmetric under $\langle \cdot, \cdot \rangle_h$, so

$$\left\langle D^{(1)}\big(\tilde{u}^{n+\frac{1}{2}}\big), \tilde{u}^{n+1} + \tilde{u}^n \right\rangle_h = 0,$$

which implies no energy contribution from the transport part.

*4.2.0.3 Numerical integral term.*
By Lemma 6, $D^{(2)}$ is negative semi-definite, hence

$$\left\langle D^{(2)}\big(I_{n+\frac{1}{2}}\big), \tilde{u}^{n+1} + \tilde{u}^n \right\rangle_h \leq 0.$$

Putting these results into (23) yields:

$$\frac{\|\tilde{u}^{n+1}\|_h^2 - \|\tilde{u}^n\|_h^2}{\tau} - \frac{1}{\Gamma(\alpha)}\left\langle D^{(2)}\big(I_{n+\frac{1}{2}}\big), \tilde{u}^{n+1} + \tilde{u}^n \right\rangle_h = 0,$$
(24)

and the last term is non-positive. Therefore,

$$\|\tilde{u}^{n+1}\|_h^2 \;\leq\; \|\tilde{u}^n\|_h^2, \quad \forall\, n \geq 0.$$

This completes the proof of the unconditional stability in the discrete $L^2$ norm. □

## 5 Convergence analysis

### 5.1 The convergence of Besse relaxation difference scheme

**Theorem 5.1** (Local truncation error). *With source terms and nonlinear effects included, each step of the Besse relaxation difference scheme has a local truncation error (LTE) on the order of*

$$\text{LTE}_m = O\big(\tau^{1+\alpha} + h^2\big),$$

*depending on whether the solution is smooth or has singular memory effects (i.e., $\alpha$-fractional).*

**Remark 2.** *When the fractional parameter $\alpha$ is involved, the time stepping may exhibit reduced convergence order in the temporal part (on the order of $\tau^{1+\alpha}$) if the solution or memory kernel introduces additional singularities. In contrast, for sufficiently smooth solutions without strong fractional singularities, the scheme often behaves as a standard second-order method in time, hence $\tau^2$.*

**Theorem 5.2** (Global error estimate). *Let $u^n$ be the exact solution and $U^n$ be the numerical solution from the Besse relaxation difference scheme. Define the global error $e^n = U^n - u^n$. Under the stability condition ($|\xi| \leq 1$) and assuming the local truncation error in Theorem 5.1, we have*

$$\|e^n\| \leq \|e^0\| + \sum_{m=0}^{n-1} \tau \|\text{LTE}_m\| = O\big(\tau^{1+\alpha} + h^2\big).$$

*Hence the second-order Besse-relaxed scheme achieves $O(\tau^2 + h^2)$ order for smooth solutions, or $O(\tau^{1+\alpha} + h^2)$ if the fractional parameter $\alpha$ dominates.*

*Proof*: We begin by noting that each time step of the scheme contributes a local truncation error $\text{LTE}_m = O(\tau^{1+\alpha} + h^2)$, as stated in Theorem 5.1. By a standard Lax-Richtmyer argument for linear (or linearized) stable schemes, the global error satisfies

$$\|e^n\| \leq \|e^0\| + \sum_{m=0}^{n-1} \tau \|\text{LTE}_m\|.$$

Since each $\text{LTE}_m = O(\tau^{1+\alpha} + h^2)$, straightforward summation over $m = 0$ to $n-1$ gives

$$\|e^n\| = O\big(\tau^{1+\alpha} + h^2\big).$$

In the case of sufficiently smooth solutions (where fractional singularities do not degrade the time accuracy), the temporal order effectively becomes $O(\tau^2)$, yielding a second-order scheme in time and space. This completes the proof. □

### 5.2 The convergence of Besse relaxation compact difference scheme

**Theorem 5.1** (Local truncation error). *For each time step of the Besse relaxation compact difference scheme, let $\text{LTE}_n$ denote the local truncation error (including Besse-type weights and fractional operator discretization). Then, under suitable regularity assumptions on the exact solution $u^n$, one has*

$$\text{LTE}_n = O\big(\tau^{1+\alpha} + h^4\big).$$

**Remark 3.** *If the fractional exponent $\alpha$ introduces strong memory effects, the temporal error term is typically $O(\tau^{1+\alpha})$. For smoother cases or small $\alpha$, one may approach $O(\tau^2)$ in time. In*

*any case, the* fourth-order compact *approach raises the spatial accuracy to O(h⁴), improving upon standard second-order schemes.*

**Theorem 5.2** (Global error estimate)**.** *Let $U^n$ be the numerical solution produced by the fourth-order compact Besse-relaxed scheme, and let $u^n$ be the exact solution. Denote the global error by $e^n = U^n - u^n$. Then, repeating the energy estimate and using the local truncation error from Theorem 5.1, we obtain*

$$\|e^{n+1}\|_h^2 \leq \|e^n\|_h^2 + C\tau \|\mathrm{LTE}_n\|_h^2 = O(\tau^{1+\alpha} + h^4),$$

*which implies*

$$\|e^n\|_h = O(\tau^{\frac{1+\alpha}{2}} + h^2).$$

*Hence this scheme achieves higher-order accuracy in both time and space.*

*Proof*: Let the error is $e^n = U^n - u^n$. By construction, $e^n$ satisfies the same discrete relation as in (22), except for an additional inhomogeneous term $\mathrm{LTE}_n$ on the right-hand side. Applying the discrete energy estimate in the same manner as (23)–(24), we get

$$\|e^{n+1}\|_h^2 \leq \|e^n\|_h^2 + C\tau \|\mathrm{LTE}_n\|_h^2.$$

From Theorem 5.1, each $\mathrm{LTE}_n = O(\tau^{1+\alpha} + h^4)$, so

$$\|e^{n+1}\|_h^2 \leq \|e^n\|_h^2 + C'\tau (\tau^{1+\alpha} + h^4).$$

A straightforward telescoping or summation argument over $n$ then shows

$$\|e^n\|_h^2 = O(\tau^{1+\alpha} + h^4),$$

thus, there is

$$\|e^n\|_h = O(\tau^{\frac{1+\alpha}{2}} + h^2).$$

Therefore, the method attains fourth-order accuracy in space and up to second-order in time or $\tau^{(1+\alpha)/2}$ if dominated by fractional effects. This completes the proof. □

## 6 Numerical experiments

In this section, we present several numerical examples to illustrate the effectiveness and accuracy of the Besse relaxation difference scheme (7)–(11) and the compact difference scheme (14)–(18). To compare their performance, we evaluate errors and convergence rates under various spatial and temporal refinements, focusing on both point-wise and overall measures of solution accuracy.

We define two types of errors relative to the exact solution $\{u_j^n\}$:

$$E_\infty(J,N) = \max_{\substack{1 \leq j \leq J \\ 1 \leq n \leq N}} \left| U_j^n - u_j^n \right|, \quad E_2(J,N) = \sqrt{h\tau \sum_{n=1}^{N}\sum_{j=1}^{J} \left( U_j^n - u_j^n \right)^2}.$$

where $h = L/(J + 1)$ is the spatial mesh size and $\tau = T/N$ is the time step. The former $(E_\infty)$ tracks the maximum point-wise error over all grid nodes and time levels, while the latter $(E_2)$ represents the root-mean-square discrepancy across the entire spatiotemporal grid.

To quantify how the error decreases as $J$ and $N$ grow, we use the following indicators:

$$\text{Rate}^x = \log_2\left(\frac{E_\infty(2J,N)}{E_\infty(J,N)}\right), \quad \text{Rate}^t = \log_2\left(\frac{E_\infty(J,2N)}{E_\infty(J,N)}\right),$$

$$\text{Rate}_x^{(2)} = \log_2\left(\frac{E_2(2J,N)}{E_2(J,N)}\right), \quad \text{Rate}_t^{(2)} = \log_2\left(\frac{E_2(J,2N)}{E_2(J,N)}\right),$$

where $\log_2(\cdot)$ estimates the order of accuracy by comparing the error when doubling the number of spatial or temporal steps. These definitions are consistent with those in (8)–(18) and enable a direct comparison of the $L^\infty$-norm and the $L^2$-norm.

In the subsequent examples, we will report both $E_\infty$ and $E_2$ under various mesh sizes $(J)$ and time steps $(N)$, along with their associated convergence rates. This allows us to assess the numerical behavior of the two proposed schemes, especially in terms of stability and achievable accuracy for different solution types. Unless stated otherwise, $J = 1024$ and $N = 4096$ are used as baseline values.

**Example 1:** Assuming the exact solution of (1)–(3) is

$$u(x, t) = \sin \pi x - \frac{t^{\alpha + 1}}{\Gamma(\alpha + 2)} \sin 2\pi x.$$

The initial condition is $u(x, 0) = \sin \pi x$, and the right-hand side term is:

$$f(x, t) = \left[\sin \pi x - \frac{t^{\alpha + 1}}{\Gamma(\alpha + 2)} \sin 2\pi x\right]\left[\pi \cos \pi x - \frac{2\pi t^{\alpha + 1}}{\Gamma(\alpha + 2)} \cos 2\pi x\right]$$

$$-\frac{t^\alpha \sin 2\pi x}{\Gamma(\alpha + 1)} - \frac{1}{\Gamma(\alpha)}\left[\frac{-\pi^2 \sin \pi x}{\alpha} t^\alpha + \frac{4\pi^2 \sin 2\pi x}{\Gamma(2\alpha + 2)} t^{\alpha + 1}\Gamma(\alpha)\right].$$

For the case in which $u_{tt}(x, t)$ exhibits a singularity at $t = 0$, we compare the Besse relaxation difference scheme and the Besse relaxation compact difference scheme by examining both the maximum-norm and discrete $L^2$-norm errors.

To validate the spatial accuracy of the second-order scheme, we fix $N = 4096$ and vary $J$, reporting maximum-norm results for $\alpha = 0.25, 0.50, 0.75$ in Table 1, alongside corresponding $L^2$-norm errors in Table 3. From these data, the scheme achieves second-order spatial convergence and a temporal rate close to $1 + \alpha$. Fig 1 illustrates that the slopes of the maximum-norm error curves align with theoretical expectations, while Fig 2 confirms a similar trend for the $L^2$-norm.

Similarly, under the same grid settings, the fourth-order Besse relaxation compact scheme attains fourth-order spatial accuracy and a temporal order near $1 + \alpha$, as evidenced by the maximum-norm results in Table 3 and the $L^2$-norm data in Table 4. This consistency is visually reinforced by Figs 3 and 4, where the error slopes match the predicted rates. These comparisons highlight the clear advantage of the compact approach in delivering higher spatial precision.

These results highlight the clear advantage of the fourth-order compact scheme in achieving higher spatial accuracy compared to the second-order scheme.Nevertheless, due to the weak singularity at $t = 0$, both methods remain limited by the fractional parameter $\alpha$ in time

**Table 1. Maximum errors and convergence orders with varying step sizes of the Besse Relaxation Difference Scheme (temporal grid number $N = 4096$ and spatial grid number $J = 1024$ fixed).**

| $\alpha$ | Spatial convergence order | | | | Time convergence order | | | |
|---|---|---|---|---|---|---|---|---|
| | $J$ | $E_\infty(h,\tau)$ | $Rate^x$ | $CPU(s)$ | $N$ | $E_\infty(h,\tau)$ | $Rate^t$ | $CPU(s)$ |
| 0.25 | 16 | 1.4697e-2 | * | 1.8286 | 32 | 7.1411e-2 | * | 2.2123 |
| | 32 | 3.8847e-3 | 1.9196 | 2.1051 | 64 | 3.0138e-2 | 1.2446 | 4.5729 |
| | 64 | 9.9756e-4 | 1.9613 | 5.5631 | 128 | 1.2705e-2 | 1.2462 | 9.4028 |
| | 128 | 2.5018e-4 | 1.9954 | 12.7636 | 256 | 5.3489e-3 | 1.2481 | 18.2695 |
| 0.50 | 16 | 1.3478e-2 | * | 1.9036 | 32 | 3.0721e-2 | * | 2.4839 |
| | 32 | 3.5510e-3 | 1.9243 | 1.9314 | 64 | 1.0882e-2 | 1.4973 | 4.3301 |
| | 64 | 9.1457e-4 | 1.9570 | 5.5885 | 128 | 3.8509e-3 | 1.4983 | 9.1678 |
| | 128 | 2.3122e-4 | 1.9838 | 13.0534 | 256 | 1.3621e-3 | 1.4994 | 18.2089 |
| 0.75 | 16 | 1.2575e-2 | * | 1.5953 | 32 | 1.2467e-2 | * | 2.2142 |
| | 32 | 3.3185e-3 | 1.9220 | 2.3458 | 64 | 3.7069e-3 | 1.7499 | 4.3942 |
| | 64 | 8.5333e-4 | 1.9594 | 5.6509 | 128 | 1.1025e-3 | 1.7495 | 8.9153 |
| | 128 | 2.1629e-4 | 1.9802 | 13.4627 | 256 | 3.2782e-4 | 1.7498 | 18.1666 |

**Table 2. $L^2$ errors and convergence orders with varying step sizes of the Besse Relaxation Difference Scheme (temporal grid number $N = 4096$ and spatial grid number $J = 1024$ fixed).**

| $\alpha$ | Spatial convergence order | | | | Time convergence order | | | |
|---|---|---|---|---|---|---|---|---|
| | $J$ | $E_2(h,\tau)$ | $Rate_x^{(2)}$ | $CPU(s)$ | $N$ | $E_2(h,\tau)$ | $Rate_t^{(2)}$ | $CPU(s)$ |
| 0.25 | 16 | 1.2247e-2 | * | 3.0049 | 32 | 6.2873e-2 | * | 4.6987 |
| | 32 | 3.1057e-3 | 1.9742 | 5.0013 | 64 | 2.5167e-2 | 1.2423 | 8.9855 |
| | 64 | 7.8621e-4 | 1.9820 | 8.2513 | 128 | 1.0708e-2 | 1.2485 | 14.4606 |
| | 128 | 1.9813e-4 | 1.9890 | 9.5106 | 256 | 4.3580e-3 | 1.2493 | 33.5423 |
| 0.50 | 16 | 1.1123e-2 | * | 1.8945 | 32 | 2.7207e-2 | * | 4.2862 |
| | 32 | 2.7820e-3 | 1.9964 | 4.2312 | 64 | 9.2458e-3 | 1.4968 | 9.1254 |
| | 64 | 6.9631e-4 | 1.9983 | 7.7035 | 128 | 3.2683e-3 | 1.4979 | 18.5481 |
| | 128 | 1.7433e-4 | 1.9981 | 9.5414 | 256 | 1.1544e-3 | 1.4990 | 37.3611 |
| 0.75 | 16 | 1.0458e-2 | * | 1.1565 | 32 | 1.0996e-2 | * | 3.4515 |
| | 32 | 2.5940e-3 | 1.9796 | 2.5688 | 64 | 2.2198e-3 | 1.7458 | 10.0195 |
| | 64 | 6.5491e-4 | 1.9835 | 4.7173 | 128 | 9.4925e-4 | 1.7473 | 16.4823 |
| | 128 | 1.6375e-4 | 1.9849 | 9.4720 | 256 | 2.9172e-4 | 1.7495 | 29.3727 |

**Table 3. Maximum errors and convergence orders with varying step sizes of the Besse Relaxation Compact Difference Scheme (temporal grid number $N = 4096$ and spatial grid number $J = 1024$ fixed).**

| $\alpha$ | Spatial convergence order | | | | Time convergence order | | | |
|---|---|---|---|---|---|---|---|---|
| | $J$ | $E_\infty(h,\tau)$ | $Rate^x$ | $CPU(s)$ | $N$ | $E_\infty(h,\tau)$ | $Rate^t$ | $CPU(s)$ |
| 0.25 | 16 | 8.8679e-3 | * | 10.5005 | 32 | 1.4178e-1 | * | 2.4483 |
| | 32 | 6.4725e-4 | 3.9365 | 12.2225 | 64 | 5.7173e-2 | 1.2437 | 5.6630 |
| | 64 | 3.9019e-5 | 3.9074 | 22.8864 | 128 | 2.3790e-2 | 1.2458 | 15.2265 |
| | 128 | 2.6827e-6 | 3.9937 | 63.6645 | 256 | 8.9867e-3 | 1.2489 | 17.3352 |
| 0.50 | 16 | 8.0739e-3 | * | 10.5968 | 32 | 2.0238e-2 | * | 2.4839 |
| | 32 | 5.5438e-4 | 3.9439 | 11.2566 | 64 | 6.8623e-3 | 1.4928 | 4.3301 |
| | 64 | 3.5528e-5 | 3.9240 | 24.8879 | 128 | 2.4859e-3 | 1.4947 | 9.1678 |
| | 128 | 2.3948e-6 | 3.9438 | 36.0921 | 256 | 8.0983e-4 | 1.4991 | 18.2089 |
| 0.75 | 16 | 7.2415e-3 | * | 15.3389 | 32 | 8.9867e-3 | * | 13.2507 |
| | 32 | 4.8743e-4 | 3.9897 | 25.5537 | 64 | 2.3810e-3 | 1.7475 | 38.4733 |
| | 64 | 2.9974e-5 | 3.9965 | 65.1153 | 128 | 7.0726e-4 | 1.7488 | 64.3375 |
| | 128 | 2.1576e-6 | 3.9939 | 98.1240 | 256 | 2.3760e-4 | 1.7494 | 78.5537 |

accuracy. Future work could explore alternative temporal strategies or relaxation refinements to mitigate this restriction and further improve performance for singular initial-value problems.

**Table 4. $L^2$ errors and convergence orders with varying step sizes of the Besse Relaxation Compact Difference Scheme (temporal grid number $N = 4096$ and spatial grid number $J = 1024$ fixed).**

| $\alpha$ | Spatial convergence order | | | | Time convergence order | | | |
|---|---|---|---|---|---|---|---|---|
| | $J$ | $E_2(h,\tau)$ | $Rate_x^{(2)}$ | CPU(s) | $N$ | $E_2(h,\tau)$ | $Rate_t^{(2)}$ | CPU(s) |
| 0.25 | 16 | 9.7789e-4 | * | 15.2859 | 32 | 1.3598e-2 | * | 6.6382 |
| | 32 | 6.2011e-5 | 3.9824 | 27.7653 | 64 | 5.7173e-3 | 1.2458 | 14.7034 |
| | 64 | 3.9079e-6 | 3.9907 | 31.6538 | 128 | 2.3790e-3 | 1.2466 | 20.3718 |
| | 128 | 2.4088e-7 | 3.9982 | 45.9277 | 256 | 1.0250e-3 | 1.2483 | 22.5638 |
| 0.50 | 16 | 8.0418e-4 | * | 21.1878 | 32 | 2.0258e-3 | * | 3.7287 |
| | 32 | 5.1028e-5 | 3.9714 | 19.7357 | 64 | 7.5510e-4 | 1.4945 | 9.3906 |
| | 64 | 3.2037e-6 | 3.9944 | 56.5537 | 128 | 2.8859e-4 | 1.4976 | 17.2667 |
| | 128 | 2.0013e-7 | 3.9980 | 24.4799 | 256 | 1.0183e-4 | 1.4991 | 25.4824 |
| 0.75 | 16 | 7.0347e-4 | * | 41.1666 | 32 | 7.7354e-4 | * | 13.7724 |
| | 32 | 4.5076e-5 | 3.9594 | 50.4728 | 64 | 2.0770e-4 | 1.7455 | 29.0614 |
| | 64 | 2.8061e-6 | 3.9642 | 66.6870 | 128 | 6.9821e-5 | 1.7470 | 41.0010 |
| | 128 | 1.7529e-7 | 3.9826 | 105.3017 | 256 | 2.0632e-5 | 1.7489 | 70.5891 |

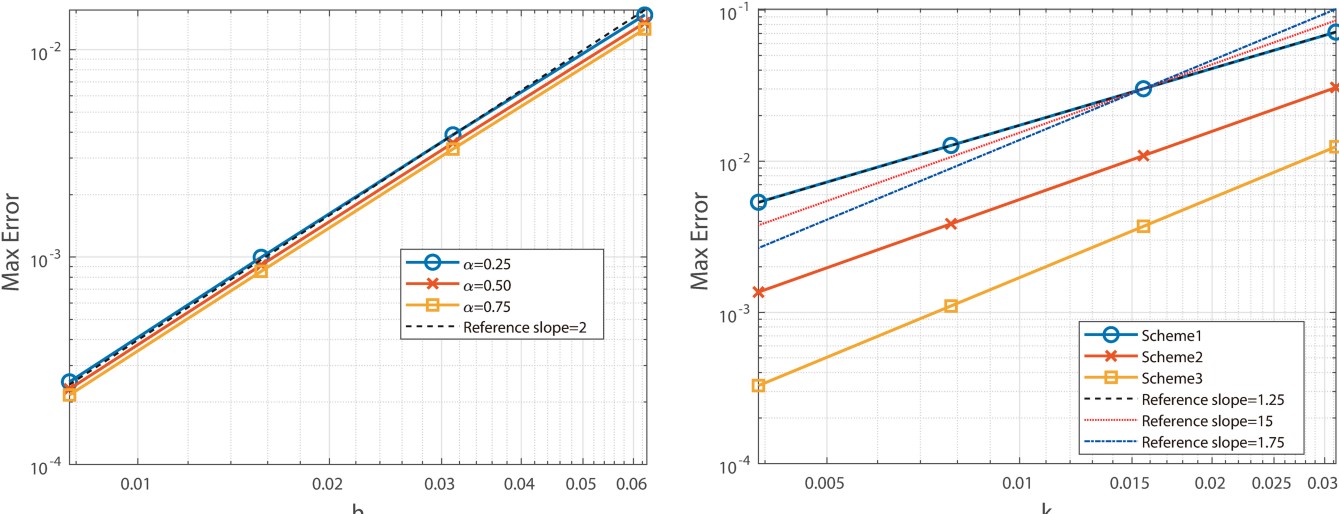

**Fig 1. Illustrates the spatial convergence order (left) and temporal convergence order (right) of the Besse relaxation difference scheme at $\alpha = 0.25$, 0.50, and 0.75.**

To further demonstrate the computational efficiency of our proposed Besse relaxation methods, we compare them against a *classical Crank–Nicolson scheme with Picard iteration* for the nonlinear algebraic equations. Specifically, we apply the Crank–Nicolson time discretization and at each time step $t_{n+1}$, we solve

$$\frac{U_j^{n+1} - U_j^n}{\tau} + \frac{1}{2}\left(U_j^{n+1} + U_j^n\right)\delta_x\left(\frac{1}{2}\left(U^{n+1} + U^n\right)\right)_j - \frac{1}{\Gamma(\alpha)}\left[\frac{1}{2}\left(I_{n+1,j}^{(\alpha)} + I_{n,j}^{(\alpha)}\right)\right] = f_j^{\left(n+\frac{1}{2}\right)},$$

via the iterative update

$$U_j^{n+1,(m+1)} = \text{PicardIteration}\left(U_j^{n+1,(m)}, U_j^n, I_{n,j}^{(\alpha)}, f_j^{\left(n+\frac{1}{2}\right)}\right),$$

until

$$\left\| U^{n+1,(m+1)} - U^{n+1,(m)} \right\|_\infty < 10^{-10}.$$

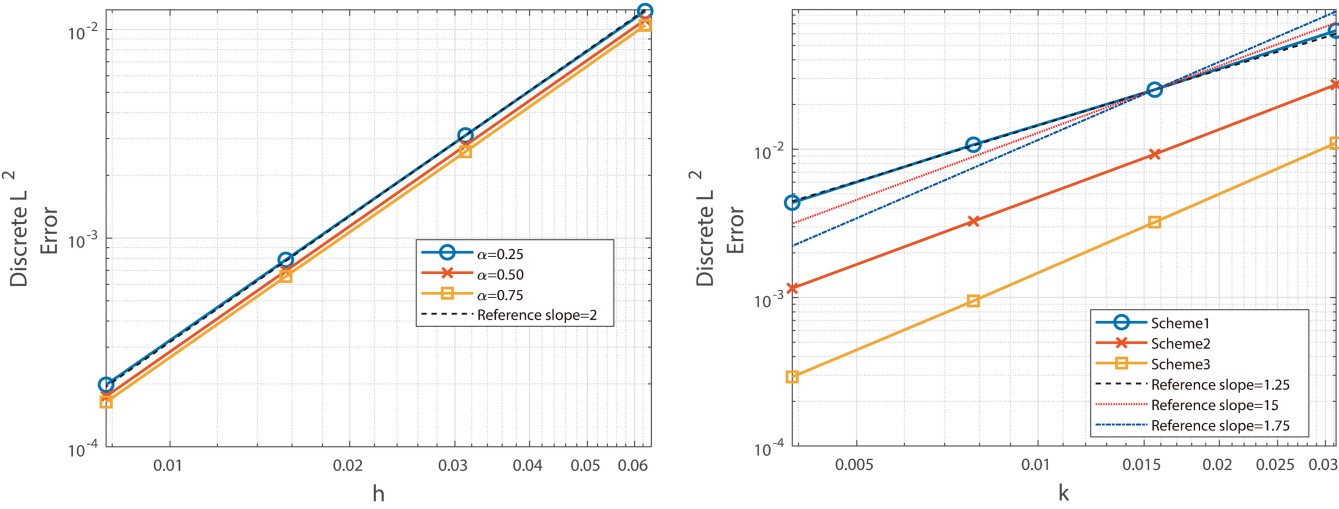

**Fig 2. Illustrates the spatial convergence order (left) and temporal convergence order (right) of the Besse relaxation difference scheme at $\alpha$ = 0.25, 0.50, and 0.75.**

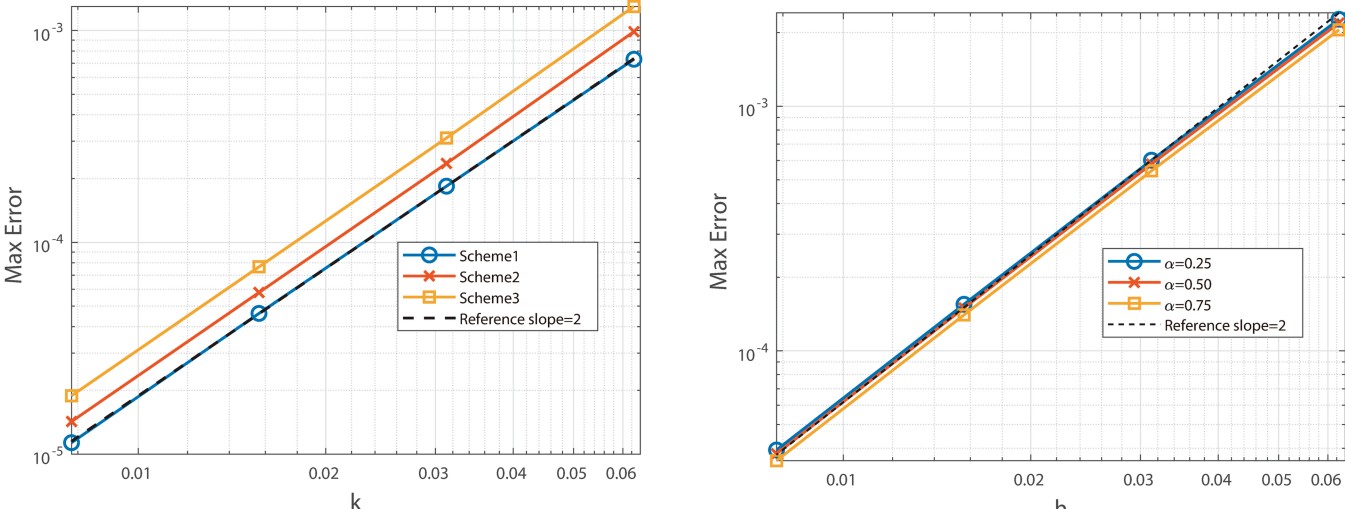

**Fig 3. Illustrates the temporal convergence order (left) and spatial convergence order (right) of the Besse relaxation difference scheme at $\alpha$ = 0.25, 0.50, and 0.75.**

All spatial differences ($\delta_x$, $\delta_{xx}$) and the fractional history terms $\{I_{n,j}^{(\alpha)}\}$ are defined consistently with our Besse relaxation scheme (see (7)–(11) and (14)–(18) for details). By running **Example 1** under identical mesh and time-step settings for both the non-relaxation Picard method and our relaxation approach, we obtain a direct CPU-time comparison that highlights how the Besse relaxation scheme can substantially reduce computational cost while preserving numerical accuracy, as shown in Tables 5 and 6.

**Example 2:** Assuming the exact solution of (1)–(3) is

$$u(x,t) = t^2 \sin\pi x,$$

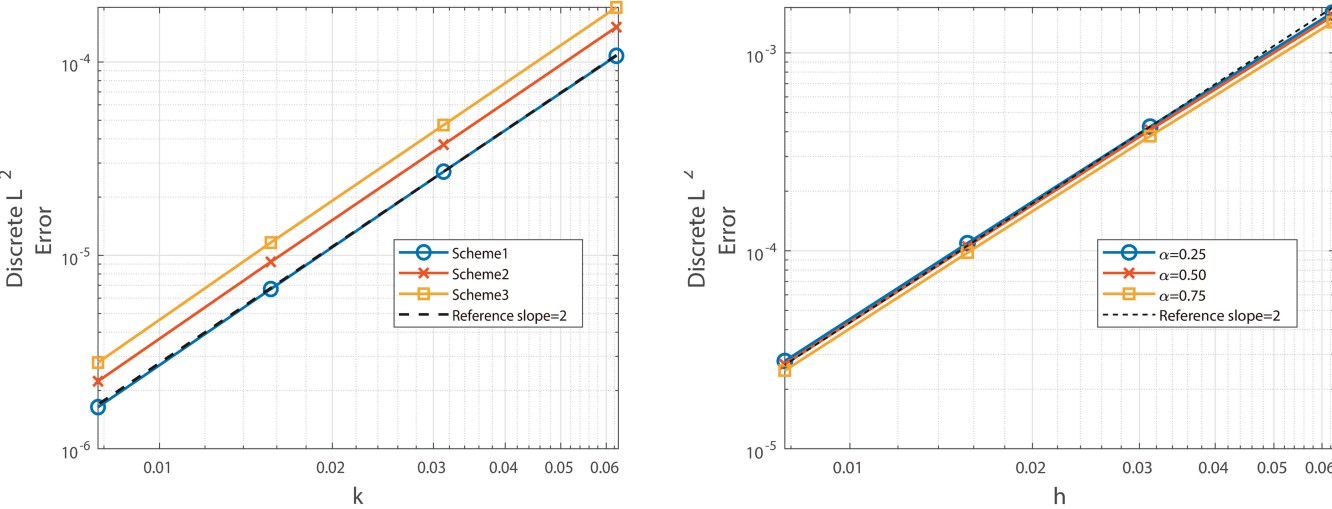

**Fig 4. Illustrates the temporal convergence order (left) and spatial convergence order (right) of the Besse relaxation difference scheme at $\alpha = 0.25, 0.50,$ and 0.75.**

**Table 5. CPU time comparison for different methods of the Besse Relaxation Difference Scheme (temporal grid number $N = 4096$ and spatial grid number $J = 1024$ fixed).**

| $\alpha$ | Besse Relaxation Difference Scheme | | | | Crank–Nicolson scheme | | | |
|---|---|---|---|---|---|---|---|---|
| | $J$ | $CPU_x(s)$ | $N$ | $CPU_t(s)$ | $J$ | $CPU_x(s)$ | $N$ | $CPU_t(s)$ |
| 0.25 | 16 | 1.8286 | 32 | 2.2123 | 16 | 3.3247 | 32 | 4.0224 |
| | 32 | 2.1051 | 64 | 4.5729 | 32 | 3.8275 | 64 | 8.3144 |
| | 64 | 5.5631 | 128 | 9.4028 | 64 | 10.1147 | 128 | 17.0960 |
| | 128 | 12.7636 | 256 | 18.2695 | 128 | 23.2065 | 256 | 33.2173 |
| 0.50 | 16 | 1.9036 | 32 | 2.4839 | 16 | 3.4611 | 32 | 4.5162 |
| | 32 | 1.9314 | 64 | 4.3301 | 32 | 3.5116 | 64 | 7.8729 |
| | 64 | 5.5885 | 128 | 9.1678 | 64 | 10.1609 | 128 | 16.6687 |
| | 128 | 13.0534 | 256 | 18.2089 | 128 | 23.7335 | 256 | 33.1071 |
| 0.75 | 16 | 1.5953 | 32 | 2.2142 | 16 | 2.9005 | 32 | 4.0258 |
| | 32 | 2.3458 | 64 | 4.3942 | 32 | 4.2651 | 64 | 7.9894 |
| | 64 | 5.6509 | 128 | 8.9153 | 64 | 11.2744 | 128 | 16.2996 |
| | 128 | 13.4627 | 256 | 18.1666 | 128 | 25.4776 | 256 | 32.9902 |

**Table 6. CPU time comparison for different methods of the Besse Relaxation Compact Difference Scheme (temporal grid number $N = 4096$ and spatial grid number $J = 1024$ fixed).**

| $\alpha$ | Besse Relaxation Compact Difference Scheme | | | | Crank–Nicolson scheme | | | |
|---|---|---|---|---|---|---|---|---|
| | $J$ | $CPU_x(s)$ | $N$ | $CPU_t(s)$ | $J$ | $CPU_x(s)$ | $N$ | $CPU_t(s)$ |
| 0.25 | 16 | 10.5005 | 32 | 2.4483 | 16 | 16.0546 | 32 | 3.8666 |
| | 32 | 12.2225 | 64 | 5.6630 | 32 | 17.9946 | 64 | 8.7123 |
| | 64 | 22.8864 | 128 | 15.2265 | 64 | 34.2098 | 128 | 23.4254 |
| | 128 | 63.6645 | 256 | 17.3352 | 128 | 98.9454 | 256 | 23.6695 |
| 0.50 | 16 | 10.5968 | 32 | 2.4839 | 16 | 16.3028 | 32 | 3.8214 |
| | 32 | 11.2566 | 64 | 4.3301 | 32 | 17.3186 | 64 | 6.6617 |
| | 64 | 24.8879 | 128 | 9.1678 | 64 | 37.2889 | 128 | 15.0243 |
| | 128 | 36.0921 | 256 | 18.2089 | 128 | 56.5267 | 256 | 27.6695 |
| 0.75 | 16 | 15.3389 | 32 | 13.2507 | 16 | 24.5980 | 32 | 21.3852 |
| | 32 | 25.5537 | 64 | 38.4733 | 32 | 37.3134 | 64 | 54.9897 |
| | 64 | 65.1153 | 128 | 64.3375 | 64 | 98.8774 | 128 | 99.9808 |
| | 128 | 98.1240 | 256 | 78.5537 | 128 | 148.9860 | 256 | 123.1518 |

The initial condition is $u(x, 0) = 0$, and the right-hand side term is:

$$f = 2t \sin \pi x - \frac{\pi}{2} t^4 \sin 2\pi x + \frac{\pi^2 \sin \pi x}{\Gamma(\alpha + 3)} t^{2+\alpha} \Gamma(3).$$

The graphical analysis of the numerical experimental results reveals that, for Example 2, the Besse relaxation difference scheme achieves a spatial convergence order of approximately two, as shown in Figs 5 and 6, while the Besse relaxation compact difference scheme attains a spatial convergence order of approximately four, as shown in Figs 7 and 8. Both schemes exhibit a temporal convergence order of approximately two. Compared to the results in Example 1, where the solution function exhibits singularity at $t = 0$, the smoothness of $u_{tt}(x, t)$

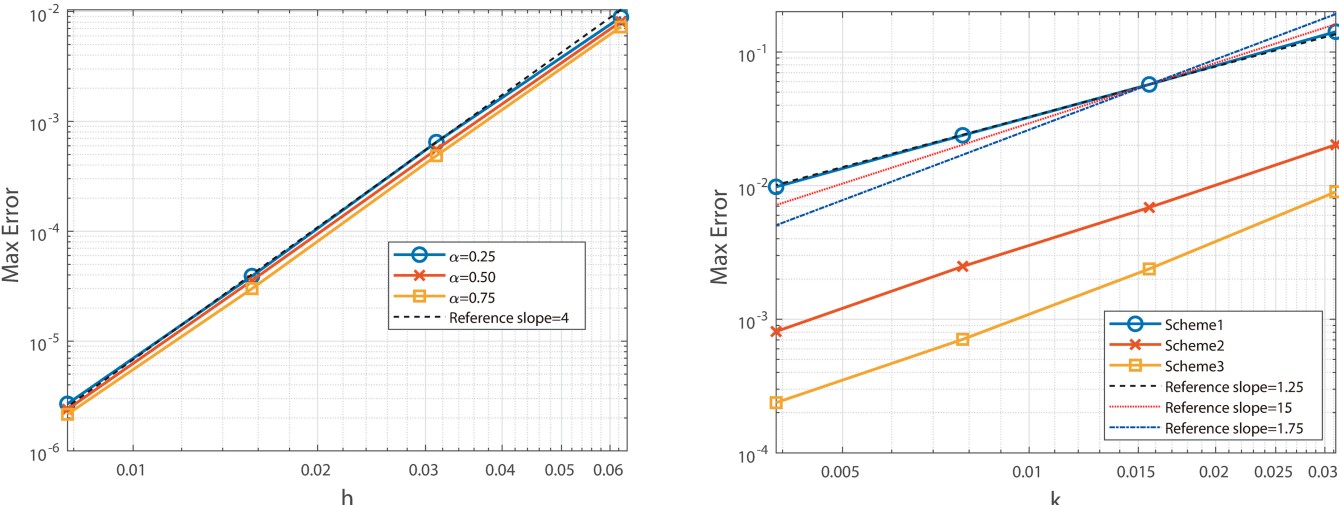

**Fig 5. Illustrates the spatial convergence order (left) and temporal convergence order (right) of Besse 358 relaxation compact difference scheme at $\alpha = 0.25, 0.50$, and $0.75$.**

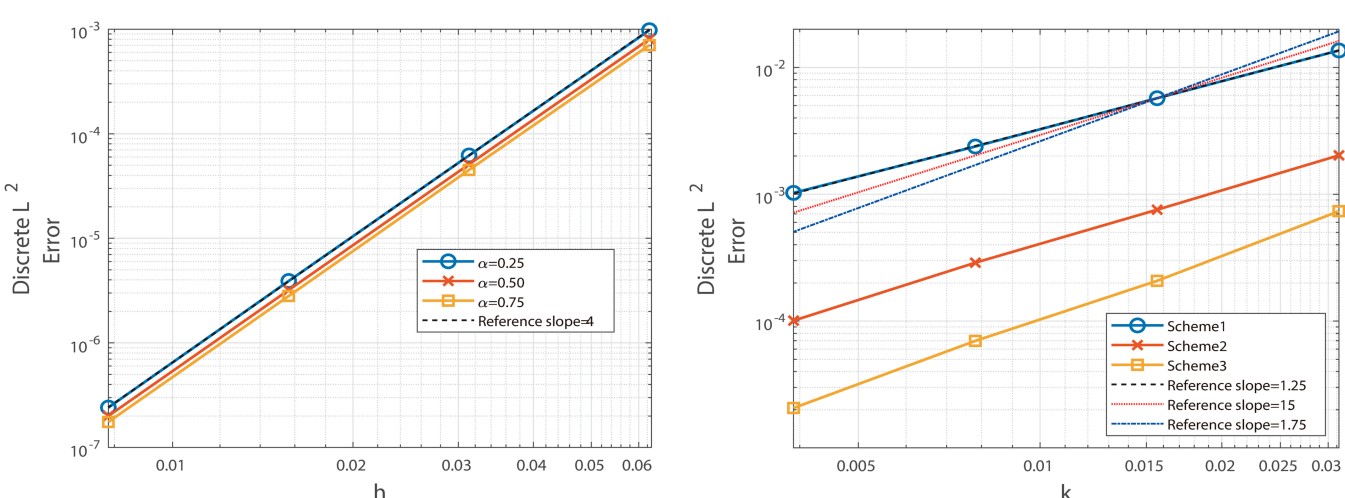

**Fig 6. Illustrates the spatial convergence order (left) and temporal convergence order (right) of Besse relaxation compact difference scheme at $\alpha = 0.25$, 0.50, and 0.75.**

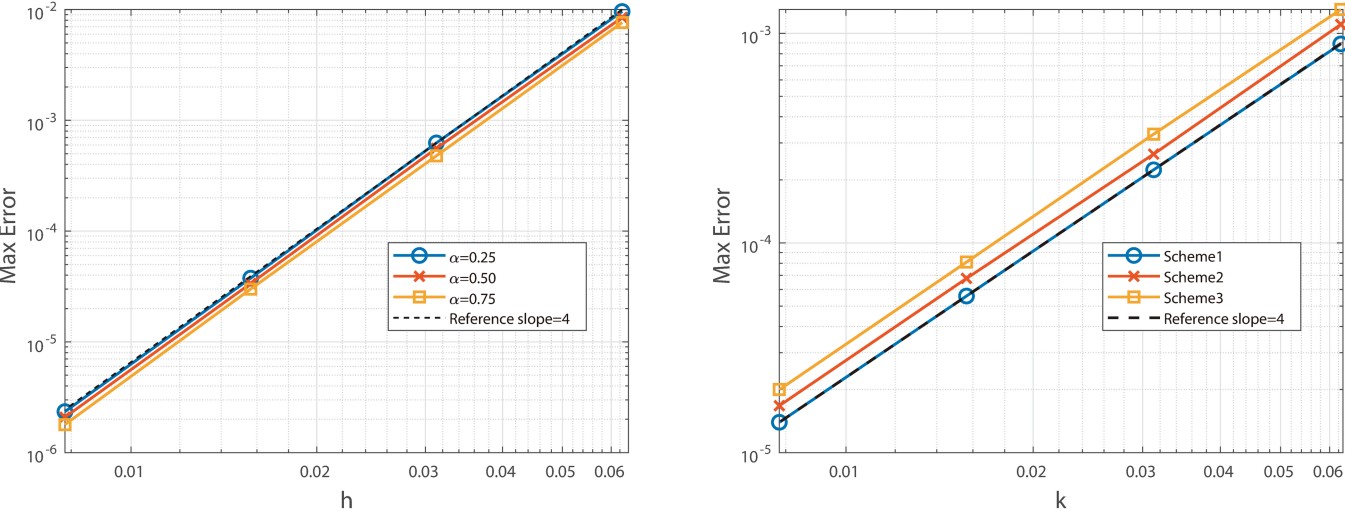

**Fig 7. Illustrates the temporal convergence order (left) and spatial convergence order (right) of Besse relaxation compact difference scheme at $\alpha$ = 0.25, 0.50, and 0.75.**

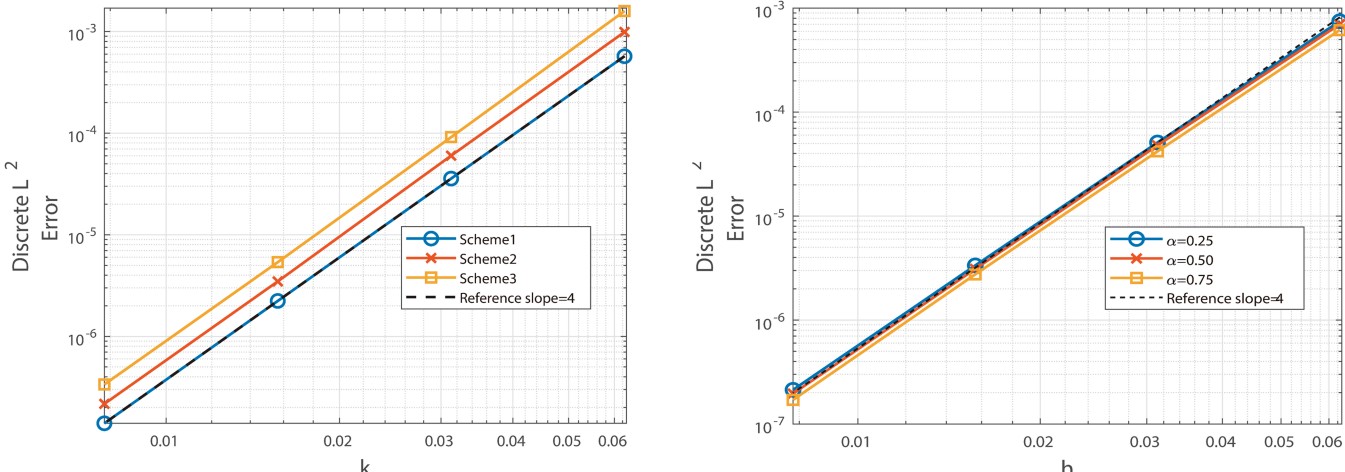

**Fig 8. Illustrates the temporal convergence order (left) and spatial convergence order (right) of Besse relaxation compact difference scheme at $\alpha$ = 0.25, 0.50, and 0.75.**

at $t = 0$ in Example 2 enables the numerical solution to achieve second-order accuracy in time. These findings are consistent with theoretical predictions, further validating the effectiveness of the proposed schemes for problems with smooth solutions.

**Example 3.** Assume the exact solution of (1)-(3) is

$$u(x,t) = t^2 x^\beta (1-x)^2, \qquad 0 < \beta < 1.$$

The initial condition is $u(x,0) = 0$, and the right–hand-side term is

$$f(x,t) = \frac{2\,t^{2-\alpha}}{\Gamma(3-\alpha)} x^\beta (1-x)^2 - t^2 \Big[ \beta(\beta-1)x^{\beta-2} - 2\beta(\beta+1)x^{\beta-1} + (\beta+2)(\beta+1)x^\beta \Big].$$

**Table 7. Maximum errors and convergence orders with varying step sizes of the Besse Relaxation Difference Scheme — Example 3 (temporal grid number $N$ = 4096 and spatial grid number $J$ = 1024 fixed).**

| $\alpha$ | Spatial convergence order | | | | Time convergence order | | | |
|---|---|---|---|---|---|---|---|---|
| | $J$ | $E_\infty(h,\tau)$ | $Rate^x$ | CPU(s) | $N$ | $E_\infty(h,\tau)$ | $Rate^t$ | CPU(s) |
| 0.25 | 16 | 1.5527e-2 | * | 1.8653 | 32 | 7.0948e-2 | * | 2.1988 |
| | 32 | 3.8890e-3 | 2.0008 | 2.1348 | 64 | 3.2177e-2 | 1.2369 | 4.5523 |
| | 64 | 9.7130e-4 | 1.9977 | 5.4899 | 128 | 1.2689e-2 | 1.2482 | 9.3401 |
| | 128 | 2.4299e-4 | 2.0011 | 12.6142 | 256 | 5.3520e-3 | 1.2473 | 15.2630 |
| 0.50 | 16 | 1.4258e-2 | * | 1.9420 | 32 | 3.1431e-2 | * | 4.5877 |
| | 32 | 3.5667e-3 | 1.9877 | 7.4647 | 64 | 1.0979e-2 | 1.4796 | 6.0913 |
| | 64 | 8.8364e-4 | 1.9992 | 15.0089 | 128 | 4.0492e-3 | 1.5013 | 10.3982 |
| | 128 | 2.2351e-4 | 2.0038 | 18.1540 | 256 | 1.4091e-3 | 1.4885 | 17.6570 |
| 0.75 | 16 | 1.3267e-2 | * | 4.6689 | 32 | 1.2554e-2 | * | 3.1377 |
| | 32 | 3.3190e-3 | 1.9970 | 8.3673 | 64 | 3.8031e-3 | 1.7478 | 7.7504 |
| | 64 | 8.5852e-4 | 1.9823 | 13.9486 | 128 | 1.0989e-3 | 1.7488 | 12.2408 |
| | 128 | 2.1477e-4 | 2.0040 | 18.3447 | 256 | 3.3216e-4 | 1.7463 | 19.8331 |

**Table 8. Maximum errors and convergence orders with varying step sizes of the Besse Relaxation Compact Difference Scheme — Example 3 (temporal grid number $N$ = 4096 and spatial grid number $J$ = 1024 fixed).**

| $\alpha$ | Spatial convergence order | | | | Time convergence order | | | |
|---|---|---|---|---|---|---|---|---|
| | $J$ | $E_\infty(h,\tau)$ | $Rate^x$ | CPU(s) | $N$ | $E_\infty(h,\tau)$ | $Rate^t$ | CPU(s) |
| 0.25 | 16 | 4.0380e-4 | * | 5.5746 | 32 | 6.9184e-3 | * | 7.0306 |
| | 32 | 2.5557e-5 | 3.9870 | 10.1318 | 64 | 2.7833e-3 | 1.2412 | 15.2235 |
| | 64 | 1.5988e-6 | 3.9903 | 24.2743 | 128 | 1.1268e-3 | 1.2447 | 20.7308 |
| | 128 | 1.0375e-7 | 3.9968 | 39.0964 | 256 | 4.5581e-4 | 1.2476 | 27.0644 |
| 0.50 | 16 | 7.4258e-4 | * | 6.6114 | 32 | 2.8733e-3 | * | 8.6402 |
| | 32 | 4.5667e-5 | 3.9879 | 13.1382 | 64 | 8.55383e-4 | 1.4877 | 14.0527 |
| | 64 | 2.9364e-6 | 3.9903 | 17.7261 | 128 | 3.2975e-4 | 1.4848 | 23.6404 |
| | 128 | 1.8430e-7 | 3.9960 | 24.4799 | 256 | 1.3174e-4 | 1.4950 | 35.1593 |
| 0.75 | 16 | 8.32675e-4 | * | 4.6591 | 32 | 1.1582e-3 | * | 14.7261 |
| | 32 | 5.2011e-5 | 4.0003 | 13.9501 | 64 | 3.3401e-4 | 1.7502 | 26.8627 |
| | 64 | 3.2507e-6 | 3.9929 | 29.7958 | 128 | 1.0869e-4 | 1.7445 | 33.8870 |
| | 128 | 2.0310e-7 | 3.9985 | 40.2632 | 256 | 3.3992e-5 | 1.7475 | 48.7473 |

*Remark.* A *q-graded* mesh with $q$ = 2 is adopted to resolve the endpoint singularity.

Tables 7 and 8 list the maximum errors and the associated convergence rates under different step sizes, with a fixed temporal grid $N$ = 4096 and spatial grid $J$ = 1024. One then refines either $h_x$ or $h_t$ to examine their respective influence on the error.

From Table 7, one observes that the spatial discretization achieves *nearly second-order accuracy*: the *Rate* column approaches 2 as $J$ doubles (i.e. $h_x$ halves), and the errors $E_s(h_x, \tau)$ decay proportionally to $h_x^2$. Turning to Table 8, the temporal convergence orders align well with theory, as the errors $E_s(h_t)$ decrease steadily with finer time steps.

Overall, these numerical results confirm that the proposed method handles PDEs with unbounded derivatives effectively: the spatial scheme is nearly second-order accurate, and the temporal scheme exhibits convergence rates in close agreement with the predictions. The good consistency between expected and observed orders underscores the robustness and efficiency of the method for such challenging problems.

## 7 Conclusions

In this paper, we presented a fourth-order Besse relaxation compact difference scheme for solving nonlinear integro-differential equations. By combining Besse-type temporal relaxation with compact finite-difference spatial discretization, our method attains high spatial

accuracy of $\mathcal{O}(h^4)$ and robust temporal convergence, which adapts effectively to both smooth solutions and those displaying singularities at $t = 0$. Numerical experiments confirm these theoretical convergence rates, showcasing the capability of scheme to capture singular behavior with minimal loss of accuracy.

Beyond one-dimensional settings, the flexibility of the Besse relaxation framework allows straightforward extension to higher-dimensional domains. The compact difference approximation naturally generalizes to multi-dimensional stencils, preserving high-order accuracy in each spatial direction. In cases where the computational domain is non-rectangular or exhibits complex geometries, standard techniques such as coordinate transformations, unstructured meshes, or local stencil adaptations can be incorporated without compromising the core algorithmic structure. Hence, the Besse relaxation approach seamlessly interfaces with advanced grid-generation methods and tailored boundary conditions, maintaining its robust stability and convergence properties.

Taken together, these features indicate that our Besse relaxation compact scheme can be adapted to a wide class of fractional-order problems in higher dimensions. The method's capacity to handle singular solutions, coupled with its high-order accuracy and computational efficiency, underscores its potential for diverse scientific and engineering applications. We anticipate that ongoing work involving multi-dimensional nonlinear integro-differential equations will further affirm the versatility and effectiveness of this approach.

## Acknowledgments

The authors would like to thank Prof. Hongbin Chen for his valuable guidance throughout this research.

## Author contributions

**Conceptualization:** Xinya Peng.

**Data curation:** Jia Zhang.

**Formal analysis:** Leiwei Li.

**Investigation:** Leiwei Li.

**Methodology:** Xinya Peng.

**Resources:** Jia Zhang.

**Supervision:** Leiwei Li.

**Writing – original draft:** Jia Zhang, Xinya Peng.

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
