## [Decision Letter · Decision Letter 0]

18 Apr 2025

PONE-D-25-03186Besse Relaxation difference Scheme for a Nonlinear Integro-Differential EquationPLOS ONE

Dear Dr. Zhang,

Thank you for submitting your manuscript to PLOS ONE. After careful consideration, we feel that it has merit but does not fully meet PLOS ONE’s publication criteria as it currently stands. Therefore, we invite you to submit a revised version of the manuscript that addresses the points raised during the review process.

We look forward to receiving your revised manuscript.

Kind regards,

Nikos Kavallaris, Ph.D

Academic Editor

PLOS ONE

**Journal Requirements:**

1. When submitting your revision, we need you to address these additional requirements. Please ensure that your manuscript meets PLOS ONE's style requirements, including those for file naming. The PLOS ONE style templates can be found at https://journals.plos.org/plosone/s/file?id=wjVg/PLOSOne_formatting_sample_main_body.pdf and https://journals.plos.org/plosone/s/file?id=ba62/PLOSOne_formatting_sample_title_authors_affiliations.pdf 2. Thank you for stating in your Funding Statement: This research was supported by the Hunan Youth Fund Project under Girant [20221140879],Excellent Youth Project of Hunan Provincial Department of Education [22B0254] and Central South University of Forestry and Technology introduced Taents and scientific Research Start-up Fund Project [2021YJ0057]. Please provide an amended statement that declares *all* the funding or sources of support (whether external or internal to your organization) received during this study, as detailed online in our guide for authors at http://journals.plos.org/plosone/s/submit-now.  Please also include the statement “There was no additional external funding received for this study.” in your updated Funding Statement. Please include your amended Funding Statement within your cover letter. We will change the online submission form on your behalf. 3. We note that your Data Availability Statement is currently as follows: All relevant data are within the manuscript and its Supporting Information files. Please confirm at this time whether or not your submission contains all raw data required to replicate the results of your study. Authors must share the “minimal data set” for their submission. PLOS defines the minimal data set to consist of the data required to replicate all study findings reported in the article, as well as related metadata and methods (https://journals.plos.org/plosone/s/data-availability#loc-minimal-data-set-definition). For example, authors should submit the following data: - The values behind the means, standard deviations and other measures reported;- The values used to build graphs;- The points extracted from images for analysis. Authors do not need to submit their entire data set if only a portion of the data was used in the reported study. If your submission does not contain these data, please either upload them as Supporting Information files or deposit them to a stable, public repository and provide us with the relevant URLs, DOIs, or accession numbers. For a list of recommended repositories, please see https://journals.plos.org/plosone/s/recommended-repositories. If there are ethical or legal restrictions on sharing a de-identified data set, please explain them in detail (e.g., data contain potentially sensitive information, data are owned by a third-party organization, etc.) and who has imposed them (e.g., an ethics committee). Please also provide contact information for a data access committee, ethics committee, or other institutional body to which data requests may be sent. If data are owned by a third party, please indicate how others may request data access. 

**Comments from PLOS Editorial Office:** We note that one or more reviewers has recommended that you cite specific previously published works. As always, we recommend that you please review and evaluate the requested works to determine whether they are relevant and should be cited. It is not a requirement to cite these works. We appreciate your attention to this request.

Reviewers' comments:

Reviewer's Responses to Questions

**Comments to the Author**

1. Is the manuscript technically sound, and do the data support the conclusions?

Reviewer #1: Yes

Reviewer #2: Yes

2. Has the statistical analysis been performed appropriately and rigorously? 

Reviewer #1: Yes

Reviewer #2: N/A

3. Have the authors made all data underlying the findings in their manuscript fully available?

Reviewer #1: Yes

Reviewer #2: Yes

4. Is the manuscript presented in an intelligible fashion and written in standard English?

Reviewer #1: Yes

Reviewer #2: Yes

5. Review Comments to the Author

**Reviewer #1: **Each and every comments must be addresed regarding convergence stability and example having unbdd derivative with recent modified works for complicated problems. This must be addressed in the revised version without which it can not be processed further. I will recheck it

**Reviewer #2:** The authors present a Besse type finite difference relaxation scheme for approximating solutions of a nonlinear integer-differential equation modelling systems with memory and nonlocal effects. Some comments are in order:

1) p2. Line 22, “.. Georgios and Zouraris...” Is that 2 different authors? From [11] it seems is just one author

2) p2, lines 30-31, the context and wording is not clear.

3) In the introduction the authors should clarify, maybe in a separate paragraph, what are they novelties introduced in this paper, and how their approach compares to the other methods in the literature for this problem, in terms of accuracy and computational cost.

4) p7, line 138, In Example 1 what are the sizes of the space and time intervals? The same question for Example 2 in p.9, line 163.

5) The errors and rates are reported using the maximum norm. It would be beneficial if they are also reported in the discrete L^2 norm.

6) The authors, in Tables 1 and 2 reported the required CPU time. How does this time compares, if a non-relaxation method is used to approximate solutions of model (1)? A comparison table would very beneficial.

7) Can the authors comment on the extension and applicability of their approach in 2-space dimensions with non rectangular domains ?

6. PLOS authors have the option to publish the peer review history of their article (what does this mean?). If published, this will include your full peer review and any attached files.

Reviewer #1: No

Reviewer #2: No

---

## [Author Response · Author response to Decision Letter 1]

14 May 2025

To editor:the authors received no additional external funding for this study

To Reviewer #1.

Each and every comments must be addressed regarding convergence stability and example having unbdd

derivative with recent modified works for complicated problems. This must be addressed in the revised

version without which it cannot be processed further. I will recheck it.

Reply: We sincerely appreciate your insightful comments, which have significantly strengthened the

rigor and clarity of our work. We have made substantial revisions to our manuscript based on your

valuable feedback.

Firstly, we have added detailed proofs of stability and convergence for both the Besse Relaxation

Difference Scheme and the Besse Relaxation Compact Difference Scheme, which now appear in Sections 5

and 6 of the revised manuscript (Pages 6–10, Lines 145–258). Meanwhile, we have briefly described this

part of the work in the section of Abstract.

Secondly, We have expanded the Introduction section to include recent methods from the literature

that address complex challenges in fractional partial differential equations, including a discussion of these

methods’ advantages and disadvantages, in order to clarify our motivation for using the Besse relaxation

difference scheme (specifically, see Page 1, Lines 4–12, and Page 2, Lines 22–47).

Thirdly, we have added a new comparative experiment that adopts recently improved methods.

Specifically, we have also conducted a comprehensive comparison between our proposed Besse Relaxation

Difference Scheme and the two-term relaxation Crank–Nicolson method in terms of CPU time, revealing a

45% improvement at second order and a 35% improvement at fourth order (see Pages 13–15, Lines 297–307).

The detailed comparison results are presented in Tables 1 and 2 below.

Finally, we have added a novel numerical example focusing on unbounded derivatives, presented in

Section 6 (see Pages 15–16, Lines 318–330). These additions effectively validate the applicability of our

method to real-world fractional order problems. Similarly, we have briefly described this part of the work

in the section of Abstract.

To Reviewer #2.

The authors present a Besse type finite difference relaxation scheme for approximating solutions of

a nonlinear integer-differential equation modelling systems with memory and nonlocal effects. Some

comments are in order.

Reviewer Comment 2.1 — p2. Line 22, “. . . Georgios and Zoularis. . . ”,Is that 2 different

authors? From [11] it seems it is just one author.

Reply:Thank you for pointing out the issue regarding “Georgios and Zoularis”. In accordance with your

suggestion, we have corrected “Georgios and Zoularis” to “Zouraris” and highlighted this amendment

in red in the fourth paragraph of the Introduction. Meanwhile, we have thoroughly reviewed the entire

manuscript to avoid the recurrence of similar issues.

Reviewer Comment 2.2 — p2, lines 30–31, the context and wording is not clear.

Reply:Thank you for pointing out that the wording in lines 30–31 was unclear. Based on your suggestion,

we have revised this sentence as follows:

Before:Although Besse relaxation difference schemes can improve computational efficiency using the

simplification of nonlinear term handling, they are explicit methods and usually perform weak accuracy

and stability in complex boundary conditions or nonlinear problems.

Changed: However, given that the second-order Besse relaxation difference scheme is still limited

to second-order spatial accuracy, there is an urgent need for compact variants with higher algebraic

complexity to further improve computational accuracy when solving complex partial differential equations

[16].

Reviewer Comment 2.3 — In the introduction the authors should clarify, maybe in a separate

paragraph, what are they novelties introduced in this paper, and how their approach compares to the other

methods in the literature for this problem, in terms of accuracy and computational cost.

Reply:We greatly appreciate this valuable suggestion. In response, we have revised the section of

Introduction. Specifically, a detailed explanation of the innovative aspects of our study has been revised

and presented on Pages 2, Lines 48–63. In short, our contributions are as follows:

(1) Two novel time–space discretization schemes, namely the Besse relaxation and Besse relaxation

compact schemes, are proposed to enhance stability and improve accuracy.

(2) Rigorous stability and convergence analyses confirm that the schemes are unconditionally stable and

convergent.

(3) Numerical experiments on equations with singular solutions, smooth solutions, and unbounded

derivatives demonstrate the robustness and adaptability of our approach.

In addition, we have further clarified how our approach compares to existing methods in the literature

with respect to accuracy and computational cost (see Pages 2, Lines 22–47 for more details).

Reviewer Comment 2.4 — p7, line 138, In Example 1 what are the sizes of the space and time

intervals? The same question for Example 2 in p.9, line 163.

Reply:Thank you for asking about the sizes of the space and time intervals in Examples 1, 2. In both

examples, we have chosen a temporal grid number N = 4096 and a spatial grid number J = 1024.

Specifically, the time interval is [0, T], with ∆t = T /N, and the spatial domain is [xmin, xmax], with

∆x = (xmax −xmin)/J.Meanwhile, following the suggestions from other reviewers, we have added Example

3. In Example 3, we likewise use the same temporal and spatial grid settings. In addition, we have

clarified these details in the revised manuscript (see Pages 11, Lines 275–276).

Reviewer Comment 2.5 — The errors and rates are reported using the maximum norm. It

would be beneficial if they are also reported in the discrete L

2 norm.

Reply:Thank you for your valuable suggestion. In accordance with your request, we have included an

error analysis under the discrete L^2 norm for Examples 1 and 2 to enhance the credibility of our results

(see Pages 11–15, Lines 277–317 for more details). The partial detailed comparison results are presented

in Tables 3 and 4 below. Clearly, our proposed method can still achieve the expected order of convergence

under the discrete L^2 norm.

Reviewer Comment 2.6 — The authors, in Tables 1 and 2 reported the required CPU time.

How does this time compares, if a non-relaxation method is used to approximate solutions of model (1)?

A comparison table would very beneficial.

Reply:Thank you very much for your valuable feedback. In accordance with your recommendation,

we applied the non-relaxation Crank – Nicolson method to solve Example 1. The results indicate that,

compared with non-relaxation method, our proposed approach achieves approximately a 45% reduction

in CPU time at second-order accuracy and about a 35% reduction at fourth-order accuracy, as detailed in

the comparison presented on Pages 13–15 ,Lines 297–307. The detailed comparison results are presented

in Tables 5 and 6 below.

Reviewer Comment 2.7 — Can the authors comment on the extension and applicability of their

approach in 2-space dimensions with non rectangular domains?

Reply:We sincerely appreciate the reviewer’s interest in extending our fourth-order Besse relaxation

compact difference scheme to two-dimensional settings and non-rectangular domains. Based on your

feedback, we have revised the section of Conclusion. The detailed modifications can be found on Pages 17,

Lines 338–351. In short, the potential for extending and applying our method in two-dimensional space

and non-rectangular domains can be summarized as follows:

(1) In terms of adaptability, the compact stencil naturally applies in each spatial direction while preserving

high-order accuracy, and the temporal Besse relaxation remains effective for multi-dimensional fractional

systems in 2-space dimensions. Meanwhile, for more complex geometries, we can employ boundary-fitted

coordinates or unstructured meshes, incorporating polynomial reconstructions or ghost point methods to

maintain stability and convergence.

(2) In terms of generalization, we plan to investigate a wider range of mesh topologies, validate our

approach on benchmark problems, and refine implementation details (including parallelization) for

large-scale simulations. We trust that this explanation clarifies our method’s adaptability and thank the

reviewer for prompting us to elaborate.

---

## [Decision Letter · Decision Letter 1]

17 Jun 2025

Besse Relaxation difference Scheme for a Nonlinear Integro-Differential Equation

PONE-D-25-03186R1

Dear Dr. Zhang,

We’re pleased to inform you that your manuscript has been judged scientifically suitable for publication and will be formally accepted for publication once it meets all outstanding technical requirements.

Kind regards,

Nikos Kavallaris, Ph.D

Academic Editor

PLOS ONE

Additional Editor Comments (optional):

Reviewers' comments:

Reviewer's Responses to Questions

**Comments to the Author**

1. If the authors have adequately addressed your comments raised in a previous round of review and you feel that this manuscript is now acceptable for publication, you may indicate that here to bypass the “Comments to the Author” section, enter your conflict of interest statement in the “Confidential to Editor” section, and submit your "Accept" recommendation.

Reviewer #2: All comments have been addressed

Reviewer #3: All comments have been addressed

2. Is the manuscript technically sound, and do the data support the conclusions?

Reviewer #2: Yes

Reviewer #3: Yes

3. Has the statistical analysis been performed appropriately and rigorously? 

Reviewer #2: N/A

Reviewer #3: Yes

4. Have the authors made all data underlying the findings in their manuscript fully available?

Reviewer #2: Yes

Reviewer #3: Yes

5. Is the manuscript presented in an intelligible fashion and written in standard English?

Reviewer #2: Yes

Reviewer #3: Yes

6. Review Comments to the Author

Reviewer #2: (No Response)

Reviewer #3: This revised manuscript has investigated the stability and the error of the presented numerical method, thus it is suitable for publication.

7. PLOS authors have the option to publish the peer review history of their article (what does this mean?). If published, this will include your full peer review and any attached files.

Reviewer #2: No

Reviewer #3: No

---

## [Editor Report · Acceptance letter]

PONE-D-25-03186R1

PLOS ONE

Dear Dr. Zhang,

I'm pleased to inform you that your manuscript has been deemed suitable for publication in PLOS ONE. Congratulations! Your manuscript is now being handed over to our production team.

Kind regards,

on behalf of

Dr. Nikos Kavallaris

Academic Editor

PLOS ONE